

# Comparative full-length transcriptome analysis by Oxford Nanopore Technologies reveals genes involved in anthocyanin accumulation in storage roots of sweet potatoes (*Ipomoea batatas* L.)

Jun Xiong[1,2], Xiuhua Tang[2], Minzheng Wei[2] and Wenjin Yu[1]

[1] Agricultural College, Guangxi University, Nanning, China
[2] Cash Crops Research Institute, Guangxi Academy of Agricultural Sciences, Nanning, China

Corresponding authors
Minzheng Wei,
Weimzheng2021@126.com
Wenjin Yu, Yuwjin163@163.com

## ABSTRACT

**Background**. Storage roots of sweet potatoes (*Ipomoea batatas L.*) with different colors vary in anthocyanin content, indicating different economically agronomic trait. As the newest DNA/RNA sequencing technology, Oxford Nanopore Technologies (ONT) have been applied in rapid transcriptome sequencing for investigation of genes related to nutrient metabolism. At present, few reports concern full-length transcriptome analysis based on ONT for study on the molecular mechanism of anthocyanin accumulation leading to color change of tuberous roots of sweet potato cultivars.
**Results**. The storage roots of purple-fleshed sweet potato (PFSP) and white-fleshed sweet potato (WFSP) at different developmental stages were subjected to anthocyanin content comparison by UV-visible spectroscopy as well as transcriptome analysis at ONT MinION platform. UV-visible spectrophotometric measurements demonstrated the anthocyanin content of PFSP was much higher than that of WFSP. ONT RNA-Seq results showed each sample generated average 2.75 GB clean data with Full-Length Percentage (FL%) over 70% and the length of N50 ranged from 1,192 to 1,395 bp, indicating reliable data for transcriptome analysis. Subsequent analysis illustrated intron retention was the most prominent splicing event present in the resulting transcripts. As compared PFSP with WFSP at the relative developmental stages with the highest (PH *vs.* WH) and the lowest (PL *vs.* WL) anthocyanin content, 282 and 216 genes were up-regulated and two and 11 genes were down-regulated respectively. The differential expression genes involved in flavonoid biosynthesis pathway include *CCoAOMT*, *PpLDOX*, *DFR*, Cytochrome P450, *CHI*, and *CHS*. The genes encoding oxygenase superfamily were significantly up-regulated when compared PFSP with WFSP at the relative developmental stages.
**Conclusions**. Comparative full-length transcriptome analysis based on ONT serves as an effective approach to detect the differences in anthocyanin accumulation in the storage roots of different sweet potato cultivars at transcript level, with noting that some key genes can now be closely related to flavonoids biosynthesis. This study helps to improve understanding of molecular mechanism for anthocyanin accumulation

in sweet potatoes and also provides a theoretical basis for high-quality sweet potato breeding.

## INTRODUCTION

Sweet potato (*Ipomoea batatas L.*) is widely planted worldwide for its high output, available management, high nutritional value and wide range of uses (*Andreason & Olaniyi, 2021*). With respect to sweet potato production, storage roots constitute the most economically important agronomic trait, which play an important role in nutrient storage and reproduction. Sweet potato cultivars distribute globally and their storage roots show different skin color (such as white, yellow, cream, red, pink and orange) and flesh color (including orange, white, yellow, cream and purple) (*Wang et al., 2018*). Sweet potato is abundant in β-carotene, carbohydrates, minerals and dietary fiber (*De Moura, Miloff & Boy, 2015*; *Sun et al., 2014*), nutrient variation of which depends to a large extent on cultivar varieties. The nutrient composition of several cultivars has been studied, including the white-fleshed sweet potato (WFSP), purple-fleshed sweet potato (PFSP) and yellow- or orange-fleshed sweet potato cultivars. The amount of ash, protein, crude fiber, total reducing sugars, and β-carotene exhibited significant difference between yellow and white sweet potato cultivars which were planted in Rwanda (*Wang et al., 2018*). Antioxidant activities of storage roots varied widely among the sweet potato cultivars, and natural colorants such as anthocyanin that is rich in purple and red-fleshed sweet potatoes inclined to related to high antioxidant activity (*Ghasemzadeh et al., 2016*; *Sun et al., 2019*). The high antioxidant effects demonstrated that red and purple fleshy sweet potatoes have potential use in the nutraceutical industry (*Wang et al., 2020*). Flavonoids including anthocyanin have been found in many plants and are conducive to plant environmental adaptation (*Kovinich et al., 2014*), fruit development (*Jaakola, 2013*; *Petroni & Tonelli, 2011*), and even human health (*Alipour, Rashidkhani & Edalati, 2016*). They accumulate in various tissues of plants at different developmental stages (*Hichri et al., 2011*), causing color variation in plant tissues such as fruits of kiwifruit (*Man et al., 2015*) and storage roots of sweet potatoes (*Li et al., 2021*). Flavonoid biosynthesis is regulated by a diverse array of exogenous and endogenous factors present in secondary metabolic pathways, which is widely researched in many plants. However, the molecular mechanism of flavonoid biosynthesis in sweet potato is not fully elucidated and few studies attempted to reveal genes related to anthocyanin accumulation in storage roots of sweet potatoes.

RNA sequencing (RNA-Seq) has been an effective method (*Uapinyoying & Goecks, 2020*) to explore novel genes and was wildly employed to study the molecular mechanisms for growth and development in plants (*Younesi-Melerdi, Nematzadeh & Pakdin-Parizi, 2020*). Oxford Nanopore Technologies (ONT) MinION as the newest third-generation sequencing technology (TGS), has obvious advantages in achieving the specific sequencing

of full-length transcripts, quantification of isoform level, and identification of complex gene structures such as alternative splicing, polyadenylation, and fusion genes (*Slatko, Gardner & Ausubel, 2018*). In this research, the transcriptome profiles of storage roots of PFSP and WFSP at different developmental stages were identified using the ONT MinION platform in order to find genes involved in anthocyanin accumulation in storage roots of sweet potatoes. The attempt is beneficial to improving molecular understanding of anthocyanin accumulation in sweet potatoes and also provides a theoretical basis for high-quality sweet potato breeding.

## MATERIALS AND METHODS

### Plant materials

Purple-fleshed sweet potato (PFSP: Guijingshu 8) and white-fleshed sweet potato (WFSP: the mutant strain 118 of Guijingshu 8) were bred by institute of Cash Crops, Guangxi Academy of Agricultural Sciences. Both cultivars were planted in the base of Guangxi Academy of Agricultural Sciences in August (Latitude: 22.853076, Longitude: 108.245977 Altitude: 55.59). The two cultivars were planted in five rows, the length was 5 m, the row spacing was 1.0 m 0.2 m, and 14 plants were planted in each row.

Samplings were conducted in 40, 50, 60, 70, 85, 95 and 110 days after planting. Three tubers from each cultivars in every stages were taken and washed with clean water. Cultivars in every stages were taken photos and arranged in two rows in Fig. S1 (PFSP in the first row and WFSP in the second row). Each tuber was divided into two parts. One was quickly sliced and put into a 2 ml centrifuge tube with forceps and frozen in liquid nitrogen for use in full-length transcriptome. The other was cut into 1 cm × 1 cm × 1 cm tubers, dried at 40 °C for 72 h, and was used for determination of anthocyanin content. A total of 1 g sweet potato powder was extracted with 0.5 mol/L citric acid buffer (pH 3) in 60 °C water bath for 2 h. The extracted liquid was cooled, 1 ml supernatant was taken and centrifuged at 12,000 rpm for 5 min. The light absorption value of the supernatant was measured at 530 nm. The detail was in related literature (*Hong, Netzel & O'Hare, 2020*).

After sampling in the whole growth period is completed, the anthocyanin content in each period is measured to understand the change trend of anthocyanin content. Samples from two periods of low and high anthocyanin content in two cultivars were taken for RNA sequencing. The corresponding groups are listed in the Table S1.

### RNA extraction, library construction, and sequencing

Total RNA was extracted from 12 samples which were divided into four groups (three replicate samples for each group).The NEBNext Poly(A) mRNA Magnetic Isolation Module was employed to enrich poly(A) mRNA to extract total RNAs from 12 root samples. We synthesize the cDNA for sequencing according to the strand-switching protocol from Oxford Nanopore Technologies. Firstly, full-length cDNA libraries were prepared from the poly(A) mRNAs by using the cDNA-PCR Sequencing kit by Oxford Nanopore (SQKPCS109) (*Yao et al., 2020*). Then, PCR was employed to amplify the cDNA for 13 to 14 cycles with specific barcoded adapters from the Oxford Nanopore PCR Barcoding kit (SQKPBK004). Finally, before loading onto a PromethION flow cell

in a MinION sequencer, the 1D sequencing adapter was synthesized to the DNA. The sequencing was running on MinKNOW. The sequencing data were loaded to the Sequence ReadArchive (SRA), National Center for Biotechnology Information (NCBI), and the accession number is PRJNA717378.

## Transcriptome analysis, lncRNA, differentially expressed transcript (DET) identification and functional annotation

Firstly, raw reads were screened by following standard: minimum average read quality score of 7 and minimum read length of 500bp. After mapping to the rRNA database, we discarded the reads corresponding to the rRNA. Next, we identified the full-length, non-chemiric (FLNC) transcripts by searching for primer at both ends of the reads. After mimimap2 mapping to the reference genome, FLNC transcripts clusters were obtained, and the consensus isoforms were screened after polishing within pinfish in each cluster (*Hu et al., 2020*).

Subsequently, the minimap2 was used to map the consensus sequences to reference genome (*Hoang et al., 2018*). The cDNA_Cupcake package was utilized to collapse the mapped reads with min-coverage of 85% and min-identity of 90%. When folding the redundant transcripts, the 5′ difference were not considered.

Power analysis of the transcriptome data was done using the RNASeqPower package. The statistical power among different experimental groups were 0.84 (PH *vs.* WH) and 0.73 (PL *vs.* WL) respectively.

Four calculation methods of CPC, CNCI, CPAT and Pfam were combined to sort non-protein coding RNA candidates from putative protein-coding RNAs in the transcripts. The minimum length and exon number threshold filtered out the putative protein-coding RNAs. LncRNA candidates were screened from transcripts according to the standard: a length of more than 200 nt and more than two exons. Then, CPC/CNCI/CPAT/Pfam were used to distinguish the protein-coding genes from the non-coding genes for further screening (*Zhang et al., 2019*).

The DESeq R package (1.10.1) was utilized to analyze the differential expression analysis of two groups. DESeq provides a statistical procedures model based on the negative binomial distribution to detect the differential expression in digital gene expression data. The Benjamini and Hochberg's approach were used to adjust the resulting *P* values (*Benjamini & Hochberg, 1995*) to control the false discovery rate. The differentially expressed genes were assigned with a FDR < 0.01 and foldchange hang found by DESeq.

Gene function was annotated by using the following databases: GO (Gene Ontology), NR (NCBI non-redundant protein sequences), KEGG (Kyoto Encyclopedia of Genes and Genomes), Pfam (Protein family), KOG/COG/eggNOG (Clusters of Orthologous Groups of proteins) and Swiss-Prot (A manually annotated and reviewed protein sequence database). Statistical analyses were performed using the SPSS 22.0 software package (IBM SPSS, Somers, NY, USA) (*Zhao et al., 2019*).

Six genes in flavonoid biosynthesis pathway were screened for validation using quantitative real-time PCR (qRT-PCR). The primers used for qRT-PCR were listed in Table S7. The LightCycler®480 II Real-Time System (LightCycler®480 II cycler, Roche,

Carlsbad, CA, USA) was utilized to perform the qRT-PCR in a 96-well plate. The thermal profile for the PCR amplification was 95 °C for 5 min, followed by 40 cycles of 10 s at 95 °C and another 40 cycles at 60 °C for 30 s. According to the instruction's protocol, all the PCR reactions were performed using the HieffTM qPCR SYBR Green Master Mix (No Rox) (Yeasen Biotech Co., Ltd., Shanghai, China). All the qRT-RCRs analyses were conducted with three technical and three biological replicates. According to the $2^{-\Delta\Delta CT}$ method, we calculated the expression level of different genes to the control (*Livak & Schmittgen, 2001*).

## Data processing and analysis
The SPSS 22.0 software package (IBM SPSS, Somers, NY, USA) was utilized to perform the statistical analyses.

# RESULTS

## Anthocyanin content variation
In this work, two widely grown sweet potato cultivars in Guangxi province (China) viz. PFSP with purple-colored skin and WFSP with red-colored skin were studied for anthocyanin content variation in storage roots. Visual inspection of the sweet potato cultivars showed that PFSP exhibited more purple pigments than WFSP (Fig. 1A). Anthocyanin content in PFSP roots showed a trend of increase as planting days increased, with the highest level of anthocyanin detected at 95 days (Fig. 1B).

Regarding anthocyanin content in WFSP roots, at first it varied little during 40-70 days, then reached to the maximum at 85 days, and finally decrease from 85 to 110 days (Fig. 1C). As suggested, the anthocyanin content in PFSP roots was evidently much higher than that of WFSP (Fig. 1).

## Full-length transcriptome sequencing analysis
ONT RNA-Seq approach (Fig. 2A) was applied for full-length transcriptome sequencing of 12 samples (PL, PH, WL and WH with three biological replicates), and subsequent transcriptomic analysis was conducted. After stringent quality checks and data cleaning, the total clean data was obtained, and the clean data of each sample reached 2.75 GB. The length of N50 ranged from 1,191 to 1,395 (Table 1).

According to the principle of cDNA sequencing, reads with primers identified at both ends are classified to be full-length sequences. The statistics of the full-length sequences obtained in this study could refer to Table S2. The minimum and maximum full-length percentage (FL %) of samples were 70.54% and 79.26%, respectively (Table S2). Lengths of multiple full-length reads were about 1.5 kb (Fig. 2B). The obtained full-length sequences were polished to acquire consensus isoforms. All the consensus transcripts (Table S3) were mapped to the reference genome through the minimap2 software for de-redundancy analysis, generating 33,709 mapped transcripts.

## Alternative splicing analysis and functional annotation of the full-length transcriptome
Precursor mRNA produced in gene transcription can be spliced in many ways. Through alternative splicing, different exons are optioned to produce corresponding mature RNAs,

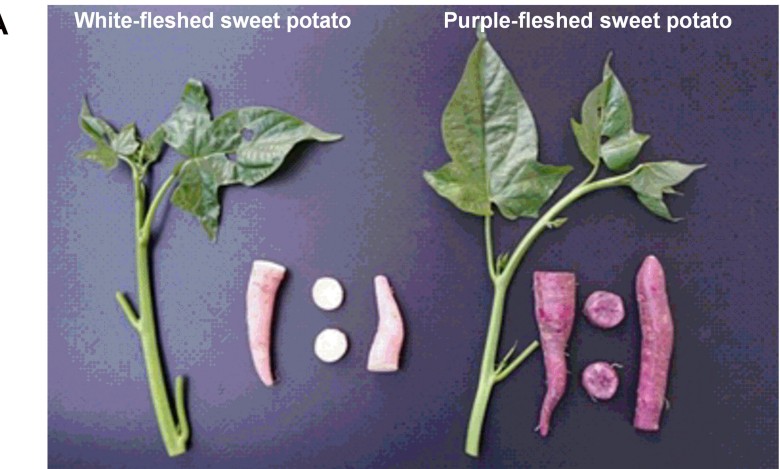

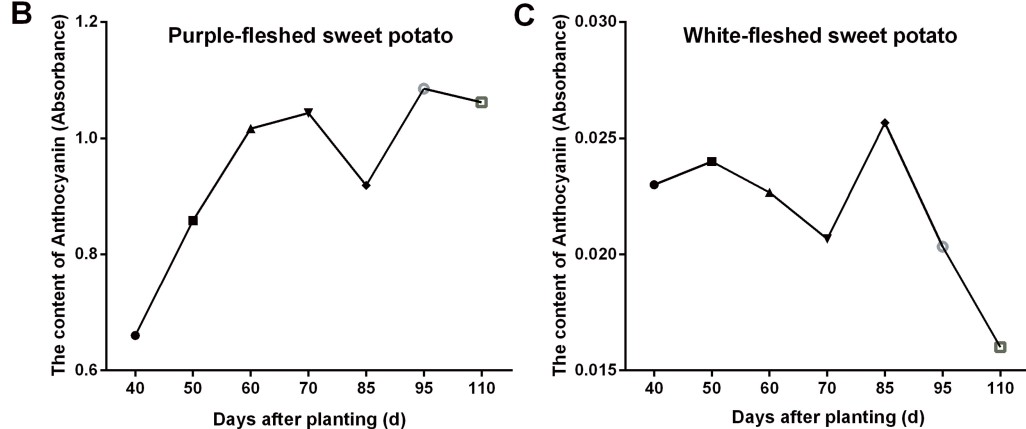

**Figure 1** **The phenotypes of sweet potatoes and anthocyanin content variation in their storage roots.**
(A) The phenotypes of PFSP and WFSP. (B) The anthocyanin profile of storage roots of PFSP during different periods. (C) The anthocyanin profile of storage roots of WFSP during different periods.

which can be translated into relative proteins representing the diversity of biological traits (*Lin & Krainer, 2019*). Based on the analysis results of Astalavista software, the full-length transcripts were statistically divided into five alternative splicing events (Fig. 2C). Figure S2 exhibited the corresponding proportion of the five alternative splicing events for each sample. Among these five alternative splicing events, intron retention events (39.8%, 11,798) were the most prominent type of alternative splicing, followed by alternative 3′ splice site (25.0%, 8427), alternative 5′ splice site (16.0%, 5393), exon skipping (14.1%, 4961) and mutually exclusive exon (5.1%, 2500), which was consistent with the findings of previous reports on other plant species (*Li et al., 2017*; *Wang et al., 2017a*).

Meanwhile, a total of 4,045 transcripts were newly found and functionally annotated by the following databases with the number of mapped reads present in the brackets: NCBI non-redundant protein sequences (NR) (4031), Gene Ontology (GO) (1842), eukaryotic Ortholog Group (KOG) (1721), and Swiss-Prot Protein Sequence (Swiss-Prot) (2483) (Table S4).

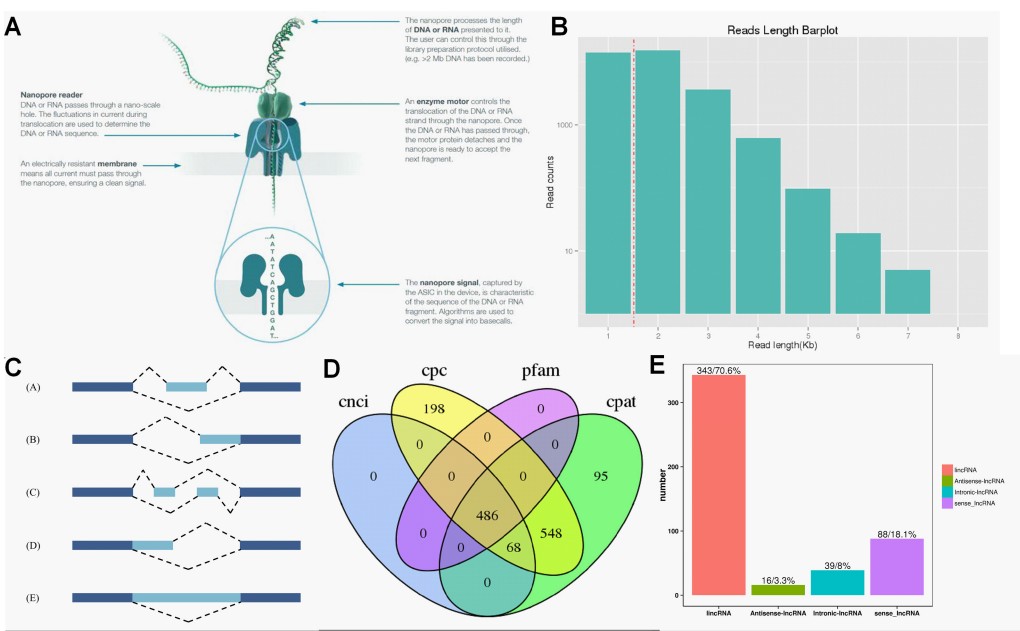

**Figure 2** **Overview of transcriptome data obtained by ONT.** (A) The principle of ONT RNA-Seq approach. (B) Length of reads. (C) The five types of alternative splicing. (D) Classification of lncRNA. (E) The number of lncRNA.

**Table 1** **Full-length transcriptome data set.**

| #FileName | ReadNum | BaseNum | N50 | MeanLength | MaxLength | MeanQscore |
|---|---|---|---|---|---|---|
| PH01 | 3447625 | 4.11E+09 | 1315 | 1191 | 11804 | Q11 |
| PH02 | 2592205 | 3.08E+09 | 1277 | 1186 | 14447 | Q10 |
| PH03 | 3386960 | 4.18E+09 | 1376 | 1233 | 12695 | Q11 |
| PL01 | 2324569 | 2.79E+09 | 1330 | 1201 | 13672 | Q10 |
| PL02 | 3173987 | 3.8E+09 | 1329 | 1196 | 11916 | Q10 |
| PL03 | 2254098 | 2.75E+09 | 1379 | 1220 | 14229 | Q10 |
| WH01 | 3380641 | 3.78E+09 | 1191 | 1118 | 9819 | Q10 |
| WH02 | 2460464 | 2.88E+09 | 1243 | 1170 | 12179 | Q10 |
| WH03 | 2471832 | 2.96E+09 | 1395 | 1198 | 11134 | Q10 |
| WL01 | 3295322 | 3.93E+09 | 1316 | 1192 | 10604 | Q11 |
| WL02 | 2288800 | 2.77E+09 | 1340 | 1210 | 11284 | Q10 |
| WL03 | 3013487 | 3.45E+09 | 1192 | 1143 | 14358 | Q11 |

## LncRNA prediction

Regulatory functions are one of the main functions of LncRNAs which are vital for post-transcription, transcription, and epigenetics (*Huang et al., 2018*). CPC analysis, CNCI analysis, CPAT, and Pfam protein domain analysis were used to predict lncRNAs from transcriptome data. A total of 486 lncRNAs were predicted by CPC, CNCI, CPAT, and Pfam software (Fig. 2D). Subsequently, lncRNAs were classified and mapped according to the reference genome annotation information. The results indicated 343 (70.6%) lncRNAs
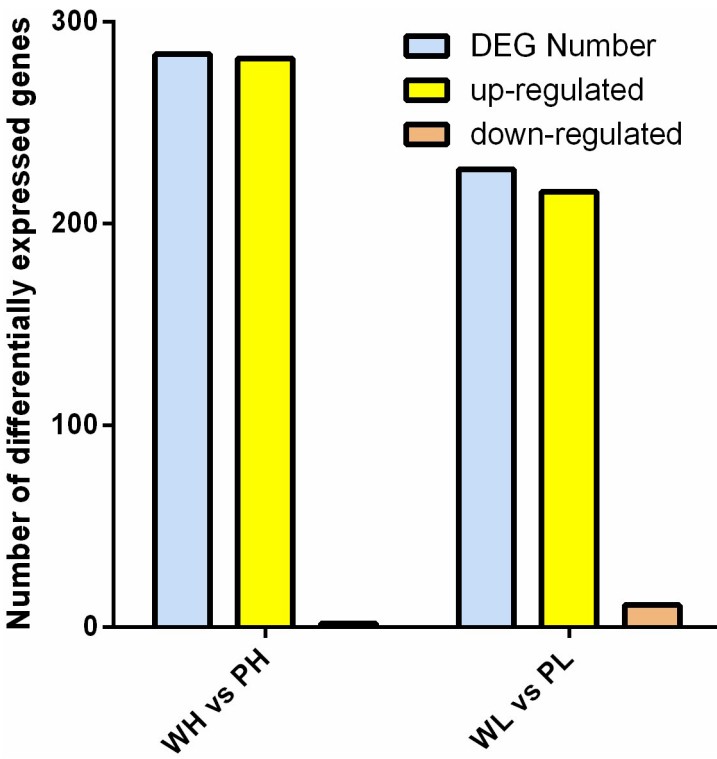

**Figure 3** **Analysis of DETs.** Histogram of DETs resulted from comparisons of transcriptome data of WH *vs.* PH and WL *vs.* PL.

were classified as lincRNA (the long non-coding RNA of intergenic region), and only 16 (3.3%) lncRNAs were sorted into antisense-lncRNA (Fig. 2E).

## Comparative analysis of differentially expressed transcripts (DET) profiling

DETs were identified between two cultivars. Data comparison of the groups PH *vs.* WH and WL *vs.* PL identified 284 and 227 DETs, respectively. Comparing the data bewteen PH and WH, 282 was up-regulated, with only two genes down-regulated. With respect to the group PL *vs.* WL, 216 and 11 genes were up-/down-regulated, respectively (Fig. 3).

GO enrichment analysis found that the DETs were significantly enriched in biological process, molecular function, and cellular component. Within biological processes, DETs were mainly involved in translation and negative regulation of endopeptidase activity. Regarding molecular function, DETs were enriched in structural constituent of ribosome. As for cellular component, DETs were mostly enriched in ribosome and cytosolic large ribosomal subunit (Figs. 4A and 4B). KEGG enrichment analysis revealed DETs between PH and WH participated in the following pathways: flavonoid biosynthesis, glutathione metabolism, flavone and flavonol biosynthesis, and phenylpropanoid biosynthesis (Fig. 4C). In terms of DEGs between PL and WL, the related significant pathways were flavonoid biosynthesis pathway, phenylpropanoid biosynthesis pathway and glutathione metabolism pathway (Fig. 4D).

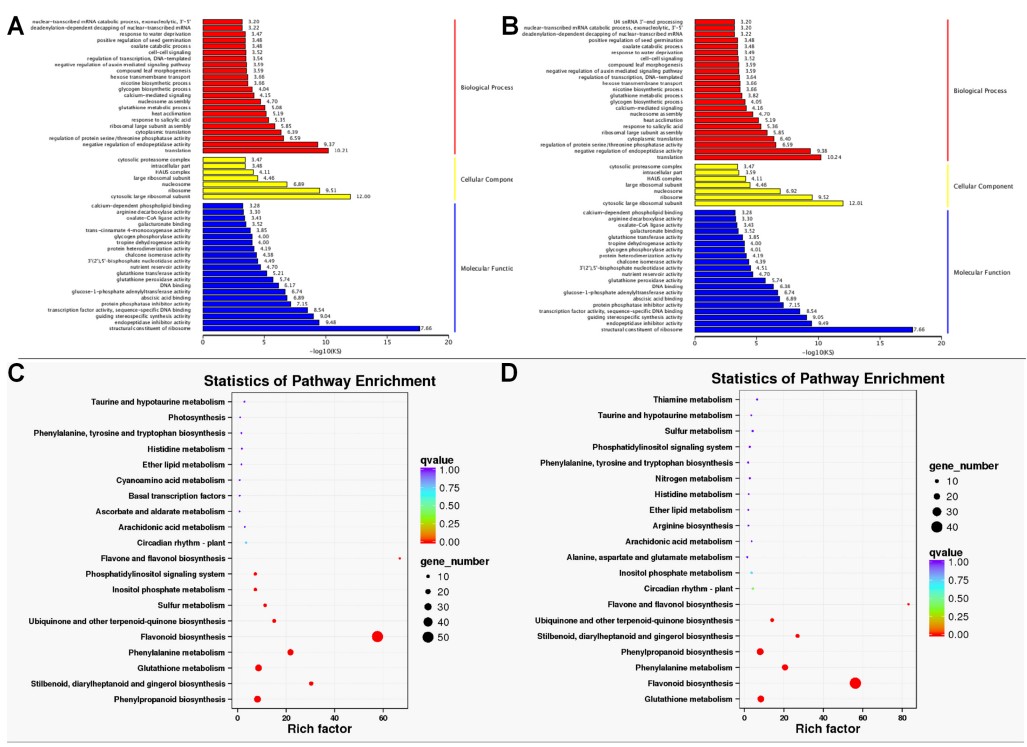

**Figure 4  GO and KEGG enrichment analysis.** GO (A) and KEGG (C) enrichment analysis of data from WH *vs.* PH and GO (B) and KEGG (D) enrichment analysis of data from WL *vs.* PL.

## Characterization of functional genes related to anthocyanin biosynthesis

Based on the transcriptome data, we identified important functional transcripts involved in flavonoid biosynthesis. Comparison the data between PH group and WH group, 51 transcripts related to anthocyanin biosynthesis were significantly up-regulated in PH group, encoding seven caffeoyl-CoA O-methyltransferase (*CCoAOMT*), five leucoanthocyanidin dioxygenase (*PpLDOX*), fourteen dihydroflavonol 4-reductase (*DFR*), ten Cytochrome P450, six chalcone isomerase (*CHI*), four chalcone synthase (*CHS*), and four 2OG-Fe(II) oxygenase superfamily. Meanwhile, bioinformatic analysis of the data between PL group and WL group showed that 40 genes involved in flavonoid biosynthesis pathway were up-regulated in PL group. The genes included five caffeoyl-CoA O-methyltransferase (*CCoAOMT*), ten dihydroflavonol 4-reductase (*DFR*), eight Cytochrome P450, six chalcone isomerase (*CHI*), four chalcone synthase (CHS), and six 2OG-Fe (II) oxygenase superfamily (Tables S5 and S6).

Notably, the genes related to anthocyanin biosynthesis including *CCoAOMT*, *PpLDOX*, *DFR*, Cytochrome P450, *CHI*, *CHS*, and 2OG-Fe (II) oxygenase superfamily were significantly up-regulated at the two developmental stages of PFSP (PL and PH) when compared with those of WFSP (WL and WH) (Tables S5 and S6).

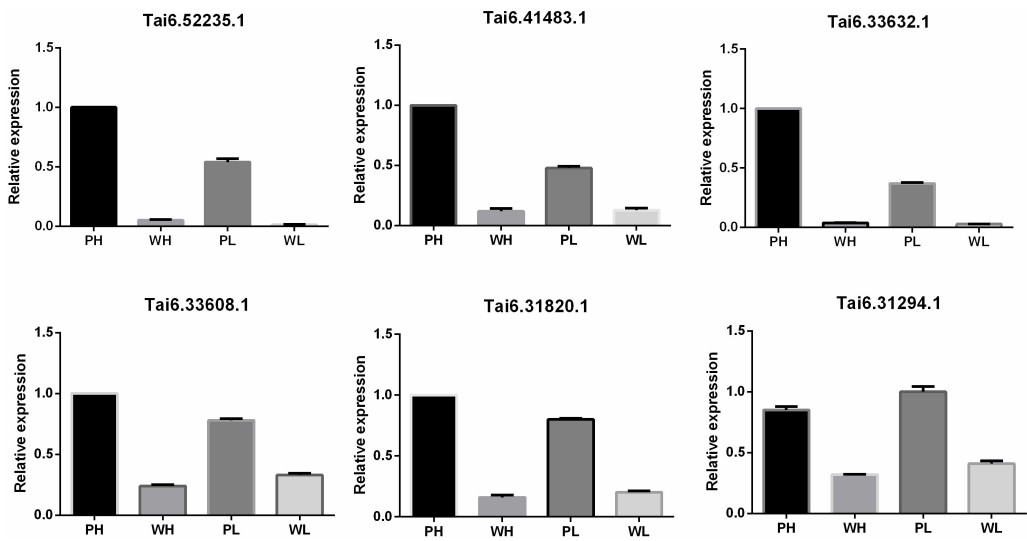

**Figure 5  Quantitative real time PCR validation for six DETs participating in the flavonoid biosynthesis pathway.**

## Validation of gene expression related to flavonoid biosynthesis pathway

To evaluate the accuracy and reproducibility of RNA-Seq data, DETs participating in the flavonoid biosynthesis pathway were selected for qRT-PCR verification. The primers used for qRT-PCR were displayed in Table S7. Pfam annotation indicated Tai6.31294, Tai6.33608, and Tai6.31820 were DETs representing cytochrome P450. Tai6.41483 and Tai6.33632 belong to the genes encoding chalcone-flavanone isomerase. Tai6.52235 was the gene related to 2OG-Fe(II) oxygenase superfamily. The qRT-PCR confirmed that the expression tendency of these 6 transcripts was highly consistent with the RNA-Seq results (Fig. 5). Thus, the transcriptome data used for DET analysis were reliable.

## DISCUSSION

Sweet potato is a special kind of anthocyanin-rich plant which benefits people' health worldwide (*Bakuradze et al., 2019*). Purple pigmentation of storage roots as one of the commonly evaluated characteristics of sweet potato varieties often arises attributed to the increase in the content and composition of anthocyanins (*Yoshimoto et al., 2001*). The researches indicated the content and composition of anthocyanins are distinct among different sweet potato varieties (*He et al., 2020*). In this work, we systematically investigated the variation in anthocyanin content among the storage roots of the two sweet potato cultivars viz. PFSP and WFSP at different developmental stages. According to the results of spectrophotometric measurement, the anthocyanin content in the root of PFSP was evidently much higher than that of WFSP at different developmental stages, suggesting that anthocyanin content variation gives rise to different pigmentation phenotypes of sweet potatoes.
In the present study, ONT RNA-Seq as one of the newest single-molecule long-read sequencing technologies was used to conduct full-length transcriptome sequencing of 12 samples obtained from tuberous roots of PFSP and WFSP. The clean data of each sample reached 2.75 GB, and the length of N50 ranged from 1,192 to 1,395 bp, which was much longer than those in previous studies using short read RNA-Seq (*Huang et al., 2012*). It is well known that alternative splicing can select and utilize alternative splice sites in the pre-mRNA through different splicing events to generate multiple mRNA transcripts from a single gene (*Feng et al., 2019*). Splicing events are involved in many physiological processes and play an important role in growth and development of plants (*Ule & Blencowe, 2019*), ONT RNA-Seq is conducive to finding different transcript isoforms of genes and providing strong evidence for alternative splicing identification. Here, based on the transcriptome data generated by ONT, we identified five main types of alternative splicing events including intron retention, alternative 3′ splice site, alternative 5′ splice site and exon skipping, which was in line with findings of previous reports on other plant species (*Bush et al., 2017*).

Inferring lncRNA functions play an important role in elucidating the roles of lncRNAs in plant growth and adaptation, including the plant host–pathogen interaction. LncRNAs can form extremely complex secondary structures, and has multi-loop-stem structures, which can elaborate pleiotropy of lncRNAs. An increasing number of researches demonstrated that secondary structures of lncRNAs were more conservative, which is essential for lncRNA functions (*Kim & Sung, 2012*). In our research, a total of 486 lncRNAs were predicted, of which 343 (70.6%) lncRNAs were classified as lincRNA, and only 16 (3.3%) lncRNAs were sorted into antisense-lncRNA.

Subsequently, DETs profiling was conducted based on the transcriptome data, indicating most of DETs were up-regulated in PFSP as compared with WFSP. KEGG enrichment analysis revealed many DETs associated with flavonoid biosynthesis pathway including *CCoAOMT*, *PpLDOX*, *DFR*, Cytochrome P450, *CHI*, *CHS*, and 2OG-Fe(II) oxygenase superfamily were significantly up-regulated in PFSP cultivars. The up-regulated DETs might execute their significant regulatory functions for anthocyanin biosynthesis in PFSP cultivars. Accumulating evidence suggests the biological processes catalysed by *CHS*, *CHI*, *F3H*, *F3H*, *F3H ly*, *DFR*, *ANS* and 3 *GT* bring out the production of different anthocyanin subgroups by modifying the molecular skeleton and/or backbone. It was reported that a much higher transcript abundance of most of the structural genes including *IbC4H*, *IbCHS*, *IbCHI*, *IbF3H*, *IbDFR*, *IbANS*, and *IbUGT* was observed in PFSP cultivars when compared to WFSP and yellow- or orange-fleshed sweet potato (*Wang et al., 2018*). Similarly, most of the structural genes involved in the anthocyanin biosynthesis pathway were up-regulated at fruit developmental stages of red pear as compared to its green color mutations (*Yang et al., 2015*). Coordinated changes in expression of *DFR*, *CHI*, and *CHS* have also been found in differently colored Chinese bayberries (*Niu et al., 2010*), grapes (*Boss, Davies & Robinson, 1996*), Arabidopsis (*Saito et al., 2013*) and other plants (*Wang et al., 2017b*). In the present study, comparative full-length transcriptome analysis by ONT revealed expression variation of important genes such as *CHS* and *DFR*, which significantly affected anthocyanin accumulation in storage roots of PFSP and WFSP. It could be inferred that transcriptional regulation of the flavonoid biosynthesis pathway based on transcriptome

data by ONT is beneficial to the accumulation of anthocyanin and other valuable flavonoids in sweet potatoes, thus improving economically agronomic traits and accelerating the breeding process.

## CONCLUSIONS

Comparative full-length transcriptome analysis based on ONT serves as an effective approach to investigate the differences in anthocyanin accumulation in the storage roots of different sweet potato cultivars at transcript level, with the notable outcome that several key genes can now be closely linked to flavonoids biosynthesis. This study helps to improve understanding of molecular mechanism for anthocyanin accumulation in sweet potatoes and also provides a theoretical basis for high-quality sweet potato breeding.

### Funding
This research was funded by the Fundamental Research Fund of Guangxi Academy of Agriculture Sciences (2021YT060) and the Guangxi Key Research and Development Plan Project (GuiKe AB16380085; GuiKe AB18221101). The funders had no role in study design, data collection and analysis, decision to publish, or preparation of the manuscript.

### Grant Disclosures
The following grant information was disclosed by the authors:
Fundamental Research Fund of Guangxi Academy of Agriculture Sciences: 2021YT060.
Guangxi Key Research and Development Plan Project: AB16380085, AB18221101.

### Competing Interests
The authors declare there are no competing interests.

### Author Contributions
- Jun Xiong performed the experiments, analyzed the data, authored or reviewed drafts of the article, and approved the final draft.
- Xiuhua Tang analyzed the data, authored or reviewed drafts of the article, and approved the final draft.
- Minzheng Wei conceived and designed the experiments, performed the experiments, analyzed the data, prepared figures and/or tables, authored or reviewed drafts of the article, and approved the final draft.
- Wenjin Yu conceived and designed the experiments, performed the experiments, analyzed the data, prepared figures and/or tables, authored or reviewed drafts of the article, and approved the final draft.

### DNA Deposition
The following information was supplied regarding the deposition of DNA sequences:
The sequencing data are available at the Sequence Read Archive (SRA), National Center for Biotechnology Information (NCBI): PRJNA717378.
## Data Availability

The raw data and code are available in the Supplemental Files.

## Supplemental Information

Supplemental information for this article can be found online at http://dx.doi.org/10.7717/peerj.13688#supplemental-information.

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
