# Peer review of "Comparative full-length transcriptome analysis by Oxford Nanopore Technologies reveals genes involved in anthocyanin accumulation in storage roots of sweet potatoes (Ipomoea batatas L.)"

_PeerJ, doi:10.7717/peerj.13688_

## Round 0.1 · original submission · Major Revisions

Dear Dr. Wei and Dr. Yu,

As you will see, our reviewers found that your work was potentially important. However, they provided several comments and suggestions to you.

Reviewer 1 had major concerns about the novelty and manuscript organization. For example, you just identified some candidate genes involved in flavonoid biosynthesis in the species, however, there was no relevant mechanistic insight. Reviewer 2 suggests that the data presentation can be improved. (S)he also had concerns about the description of the materials and methods. For example, clarification of sampling days and software version and their parameters. The both reviewers point out that the manuscript needs language editing by proficient English speakers.

I would like to ask you to address or to respond with reasons not to follow the suggestion made by the reviewers.

Best regards,
Atsushi Fukushima

Reviewer 1 ·

Basic reporting

no comment

Experimental design

no comment

Validity of the findings

no comment

Additional comments

The manuscript entitled “Comparative full-length transcriptome analysis by Oxford Nanopore Technologies reveals genes involved in anthocyanin accumulation in storage roots of sweet potatoes (Ipomoea batatas L.)” presented comparative RNA-seq analysis between two different sweet potatoes as well as pigment detection, based on which the author found out some candidate genes involved in flavonoid biosynthesis, aiming to provide basic information for future breeding. The overall research is relatively standardized, but lack of new ideas. Finally the author presented prosaic manuscript, in which the method and analysis results were regular without special points. What need to extensively modify were below.
Major point:
--Paragraph 2 in Introduction section, the author spent large content to introduce the RNA sequencing and focused on the 3rd generation sequencing technology, which was not necessary. The purpose of this research is to discover relevant genes involved in flavonoid biosynthesis between two different sweet potato cultivars simply named PFSP and WFSP rather than explore the development of RNA-seq technology, so the descriptions in introduction section should focus on the purpose of study rather than others. The same problem appeared in the discussion section. The author need to revised or removed. Similarly, in Figure 2, it was meaningless to introduce the ONT RNA-seq approach (Figure 2B) and the types of alternative splicing (Figure 2D), which had nothing to do with the aim of your research.
--Line 92-94, “In this research, the transcriptome profiles of storage roots of PFSP and WFSP at different developmental stages were identified using the ONT MinION platform in order to reveal the molecular mechanisms for anthocyanin accumulation in the two sweet potato cultivars”, molecular mechanism in this sentence is an overstatement because the study in only a preliminary exploration for anthocyanin accumulation in compared two cultivars. The author just provided some candidate genes involved in flavonoid biosynthesis, no relevant mechanism research were involved in this research. Therefore, the author need to revise the inappropriate wording statement throughout manuscript.
--In anthocyanin detection, it is not enough that the author only mentioned the method of UV-visible spectrophotometric measurements. The specific and detailed steps and parameters should be described in manuscript. How to extract anthocyanin from samples? Which wavelength was used in measurements?
--Figure 1, I am confused why the author separated the anthocyanin content of PFSP and WFSP into two figures. Putting Figure 1B and C together could more intuitively reflect the difference of anthocyanin content of PFSP and WFSP. Additionally, the anthocyanin content corresponded to the samples of seven periods and the content presented a dynamic change trend, why the author did not place seven corresponding phenotype of PFSP and WFSP? This would be more visible.
--Most of parts in discussion section were the simply repeated presentation of the results from the article, without the author’s own views or introducing new perspectives. The author should make extensive modifications in discussion section.
--The English language need to improve.
Minor point:
--Line 63, “The amount of ash...”, what is the meaning of ash here.
--Line 118, “firstly”, should be “Firstly”.
--Line 123, “ ...1D sequencing adapter...”, what is the meaning of 1D?
--Line 124, “ loadingto”, should be “loaded to”
--Line 128, “...minimum average read quality score of 7...”, what is the criteria for defining the read quality score?
--Line 218-219, “With respect to the group PL vs. WL 216 genes were up-regulated and 11 genes were down-regulated”, should be “With respect to the group PL vs. WL, 216 and 11 genes were up-/down-regulated, respectively”
--Line 231-233, the author identified transcripts based on the transcriptome data between PH and WH comparison, which meant the candidate gene screening were depended on the different expression of genes between different comparison. 3rd sequencing mainly provided specific sequence information while 2nd sequencing provided differential expression. So I want to know whether the author have perform 2nd transcriptome analysis.
--In conclusion section, the last sentence is the significance of your research rather than your study conclusion.

Reviewer 2 ·

Basic reporting

no comment

Experimental design

no comment

Validity of the findings

no comment

Additional comments

see attached

Annotated reviews are not available for download in order to protect the identity of reviewers who chose to remain anonymous.

---

## Round 0.2 · Major Revisions

Dear authors,

Thank you for your revision. However, Reviewer 2 was not satisfied with the revised manuscript. Would you please revise it according to the comments by the reviewer?

Best regards

Reviewer 1 ·

Basic reporting

no comment

Experimental design

no comment

Validity of the findings

no comment

Additional comments

The author have addressed my concerns carefully in the revised version that could be published.

Reviewer 2 ·

Basic reporting

no comment

Experimental design

no comment

Validity of the findings

no comment

Additional comments

I don’t think the authors have properly revised the manuscript. Not reflected in the main text. Also, there are still many mistakes.

Annotated reviews are not available for download in order to protect the identity of reviewers who chose to remain anonymous.

---

## Round 0.3 · accepted · Accept

Dear authors,

Thank you for the revision.

Best regards

For instance:

line 150 "foldchange hang" - something needs to be fixed

Reviewer 2 ·

Basic reporting

no comment.

Experimental design

no comment

Validity of the findings

no comment

Additional comments

I think the authors have properly revised the manuscript based on the reviewer's comments.